# lra: A long read aligner for sequences and contigs

**Jingwen Ren**, **Mark J. P. Chaisson** *

Department of Quantitative and Computational Biology (QCB), University of Southern California, Los Angeles, California, the United States of America

* mchaisso@usc.edu

**Data Availability Statement:** HG002 hifisam assembly data can be downloaded using links: https://zenodo.org/record/4393631/files/NA24385.HiFi.hifiasm-0.12.hap1.fa.gz?download=1 https://zenodo.org/record/4393631/files/NA24385.HiFi.hifiasm-0.12.hap2.fa.gz?download=1 The HG002

## Abstract

It is computationally challenging to detect variation by aligning single-molecule sequencing (SMS) reads, or contigs from SMS assemblies. One approach to efficiently align SMS reads is sparse dynamic programming (SDP), where optimal chains of exact matches are found between the sequence and the genome. While straightforward implementations of SDP penalize gaps with a cost that is a linear function of gap length, biological variation is more accurately represented when gap cost is a concave function of gap length. We have developed a method, lra, that uses SDP with a concave-cost gap penalty, and used lra to align long-read sequences from PacBio and Oxford Nanopore (ONT) instruments as well as de novo assembly contigs. This alignment approach increases sensitivity and specificity for SV discovery, particularly for variants above 1kb and when discovering variation from ONT reads, while having runtime that are comparable (1.05-3.76×) to current methods. When applied to calling variation from *de novo* assembly contigs, there is a 3.2% increase in Truvari F1 score compared to minimap2+htsbox. lra is available in bioconda (https://anaconda.org/bioconda/lra) and github (https://github.com/ChaissonLab/LRA).

## Author summary

Any two human genomes will have sequence differences across multiple scales: from single-nucleotide variants to large gains, losses, or rearrangements of DNA called structural variants. Long-read single-molecule sequencing has been shown to help discover structural variation because the reads span across the entire variant. The computational problem for discovering a structural variant is to find the optimal alignment of the read to the genome with gaps that accurately reflect the variant. Here we demonstrate a method, lra, that uses an efficient implementation of concave-cost alignment for structural variant discovery using long reads. On standardized benchmark data, we show that structural variant discovery is improved for multiple combinations of variant detection algorithms and long-read sequence using alignments generated by lra compared to existing methods. Finally, we show that it is possible to use lra to accurately discover a complete spectrum of structural variants using de novo assemblies constructed from long-read sequence data. This implies a future model of comparative genomics where variants are discovered only by comparing de novo assemblies and not a comparison of reads against a reference.

ONT reads are available in NCBI database under BioProject accession number PRJNA678534. The HG002 HiFi reads can be downloaded using links: https://s3-us-west-2.amazonaws.com/human-pangenomics/index.html?prefix=NHGRI_UCSC_panel/HG002/hpp_HG002_NA24385_son_v1/PacBio_HiFi/19kb/m64011_190714_120746.Q20.fastq https://s3-us-west-2.amazonaws.com/human-pangenomics/index.html?prefix=NHGRI_UCSC_panel/HG002/hpp_HG002_NA24385_son_v1/PacBio_HiFi/19kb/m64011_190728_111204.Q20.fastq https://s3-us-west-2.amazonaws.com/human-pangenomics/index.html?prefix=NHGRI_UCSC_panel/HG002/hpp_HG002_NA24385_son_v1/PacBio_HiFi/20kb/m64011_190830_220126.Q20.fastq https://s3-us-west-2.amazonaws.com/human-pangenomics/index.html?prefix=NHGRI_UCSC_panel/HG002/hpp_HG002_NA24385_son_v1/PacBio_HiFi/20kb/m64011_190901_095311.Q20.fastq https://s3-us-west-2.amazonaws.com/human-pangenomics/index.html?prefix=NHGRI_UCSC_panel/HG002/hpp_HG002_NA24385_son_v1/PacBio_HiFi/25kb/m64011_190712_225711.Q20.fastq https://s3-us-west-2.amazonaws.com/human-pangenomics/index.html?prefix=NHGRI_UCSC_panel/HG002/hpp_HG002_NA24385_son_v1/PacBio_HiFi/25kb/m64011_190726_220327.Q20.fastq The HG002 CLR reads can be downloaded using the following links: https://s3-us-west-2.amazonaws.com/human-pangenomics/index.html?prefix=NHGRI_UCSC_panel/HG002/hpp_HG002_NA24385_son_v1/PacBio_CLR/WUSTL_SV-HG002-CLR/1_A01/m64043_191010_174437.subreads.bam https://s3-us-west-2.amazonaws.com/human-pangenomics/index.html?prefix=NHGRI_UCSC_panel/HG002/hpp_HG002_NA24385_son_v1/PacBio_CLR/WUSTL_SV-HG002-CLR/3_C01/m64043_191012_102127.subreads.bam Variant calls, and a set of curated inversions are available at: https://figshare.com/articles/dataset/lra-supplemental-HG002-SV_vcf_tar_gz/13238717.

**Funding:** M.J.P.C. is supported by NHGRI U24HG007497 and NHGRI 1U01HG010973. The funders had no role in study design, data collection and analysis, decision to publish, or preparation of the manuscript.

**Competing interests:** The authors have declared that no competing interests exist.

This is a *PLOS Computational Biology* Methods paper.

## Introduction

Studies of genetic variation often begin by aligning sequences from a sample back to a reference genome, and inferring variation as differences in the alignment. Long read, single molecule sequencing (LRS) is becoming established as a routine approach for sequencing genomes. The two technologies that produce LRS technologies, Pacific Biosciences (PacBio) and Oxford Nanopore (ONT) generate reads over 50kb at error rate 15% or less. Aligning these sequences is a computationally challenging task for which several methods are available including minimap2, ngmlr, and BLASR [1–3]. They are demonstrated to be quite fast and accurate, but have limitations, particularly when there are large sequence differences between the read and the reference. This problem is amplified in complex, repetitive regions such as variable-number tandem repeats, that only make up 3% of the human genome, but account for nearly 70% of observed structural variation: insertions and deletions at least 50 bases (SV), and in larger SV [4].

A common approach for mapping LRS reads is seeding and chaining, where an approximate alignment is formed based on a subset (chain) of exact matches between the sequence and the genome. The exact matches may be found using various data structures including variable-length matches using a BWT-FM index or suffix arrays [3], and minimizer based indexing [1]. The chaining algorithm used by BLASR uses a linear cost gap function for sparse-dynamic programming (SDP) [5]. While the chaining algorithm is efficient, it has long been known that linear-cost gap functions do not accurately reflect biological variation [6], and has been shown to decrease sensitivity for detecting SV from LRS alignments [2].

Both the minimap2 and ngmlr aligners use gap penalties that are a concave function of gap length, and are demonstrated to be quite effective for mapping LRS reads across SV with alignment gaps that reflect biological variation. In minimap2, a heuristic algorithm is used for chaining, while ngmlr adopts a Smith-Waterman-like dynamic programming algorithm. An exact solution to sparse dynamic programming with a concave gap (CG-SDP) function exists [7], and is slightly less efficient than linear-cost SDP. However, as presented, the algorithm requires asynchronous processing and has never been implemented for sequence alignment in computational biology. Furthermore, the algorithm does not take into account genome arrangements such as inversions.

An additional challenge for variant discovery is efficient alignment of contigs assembled from LRS that now have contiguity on par with the initial release of the human genome [8–10]. Existing methods exist to align whole genomes, but do not implement concave gap penalties [11], are low-resolution [12], or split alignments across large variants [1]. To explore the application of the exact solution of seed chaining sparse dynamic programming with a concave gap function to read and assembly alignment, we developed an alignment method—lra. We have simplified the implementation of the CG-SDP algorithm [7], and extended to allow for inversions and translocations. Finally, we demonstrate that this approach can improve SV discovery while having a runtime on par with state of the art methods.

## Results

We compared alignment metrics and variant discovery on simulated data sampling from the human genome build GRCh38, and real sequencing data from HG002 including three

**Table 1. Performance of alignment methods.**

| | | HG002 HiFi | HG002 CLR | HG002 ONT | HG002 HiFi hifiasm hap1 | HG002 HiFi hifiasm hap2 |
|---|---|---|---|---|---|---|
| lra | runtime | 942m | 7428m | 1528m | 105m | 90m |
| | memory | **12.01G** | 16.46G | **13.96G** | 31.10G | 28.66G |
| | # of mapped reads | 1.84M | 4.95M | 1.37M | 396 | 411 |
| | # of mapped bases | 35.85Gb | 86.82Gb | 25.45Gb | 2.87Gb | 2.98Gb |
| minimap2 | runtime | **890m** | **1973m** | **1358m** | **100m** | **82m** |
| | memory | 19.04G | 18.49G | 22.88G | **20.95G** | **21.90G** |
| | # of mapped reads | **1.85M** | **5.06M** | 1.41M | **492** | **496** |
| | # of mapped bases | **36.20Gb** | **87.36Gb** | 25.73Gb | **2.92Gb** | **3.01Gb** |
| ngmlr | runtime | 5087m | 33047m | 8862m | - | - |
| | memory | 13.6G | **14.67G** | 17.00G | - | - |
| | # of mapped reads | 1.78M | 4.84M | **2.07M** | - | - |
| | # of mapped bases | 34.59Gb | 83.75Gb | 24.22Gb | - | - |

Each dataset was aligned to GRCh37 by all methods with 16 threads, with—indicating a software crash. Total CPU time is reported. The optimal values in each class are given in bold. The HiFi hifiasm assembly is a haplotype-resolved *de novo* assembly of HG002 using HiFi reads, with haplotype N50 values of 50.55Mb and 42.92Mb.

sequencing datasets: PacBio consensus reads (HiFi), PacBio single-pass reads (CLR), and ONT reads, as well as on contigs from a haplotype-resolved *de novo* assembly [10]. The simulated data were mapped to GRCh38 to evaluate mapping accuracy with a genome that includes centromeric sequences. All real data were mapped to build GRCh37 to use curated variants for accuracy analysis [13]. The PacBio HiFi data are characterized by high-accuracy reads (>99%) with an average length of 19kb, compared to the CLR reads that have an accuracy around 80% and an average read length of 21kb. The ONT data: 88% accuracy and 1.2kb average read length. The distribution of the read lengths are shown in S1 and S2 Figs. All read datasets are above 40× coverage of a human genome.

The alignment results and structural variant callsets were compared to those generated by minimap2 and ngmlr. When computing SAM formatted alignments with minimap2, the lra and minimap2 runtime are comparable. lra and minimap2 reach similar speed on HiFi and ONT data, although all are within a factor of 1.12. minimap2 is 3.7 times faster than lra on CLR dataset (Table 1). lra is 4–6 times faster than ngmlr. The difference of bases aligned by lra and minimap2 are within 0.06–1%. We compared the mapping accuracy of lra, minimap2 and ngmlr on simulated HiFi, CLR data for read lengths between 5–50kb and ONT data for read lengths between 1–50kb. HiFi and CLR reads were simulated by PBSIM (https://github.com/pfaucon/PBSIM-PacBio-Simulator) and ONT data were simulated by alchemy2, which is distributed with lra source. We used *paftools.js mapeval* to evaluate the mapping accuracy across three overlap percentages (10%, 40%, 70%) to measure the change of mapping accuracy with overlap percentage evaluation metric. The mapping accuracy of lra for reads with mapping quality at least 20 is 99.89%, 99.97% and 99.97% for HiFi, CLR and ONT respectively, when a 40% overlap with the simulated interval is required for a correct alignment (Fig 1). As the overlap percentage goes up from 10% to 70%, the mapping accuracy of lra alignment decreases 0.1%, 0.09%, 0.03% for HiFi, CLR and ONT at mapqv 20 due to differences in lengths of sequences aligned in repetitive regions. In lra, a second local minimizer index is used to refine chained anchors and improve alignments (Methods). We evaluated the mapping accuracy of lra without the step of refining by local minimizers (S1 Text). For simulated HiFi data, the mapping accuracy has almost no difference on all 10%, 40%, 70% overlap percentage metrics,

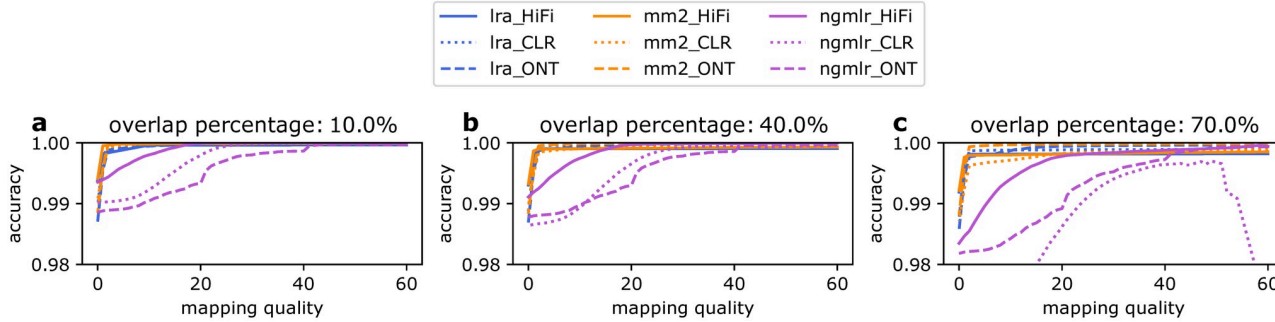

**Fig 1. Mapping accuracy.** Mapping accuracy of lra, minimap2 and ngmlr on simulated HiFi, CLR reads for lengths between 5–50kb and ONT reads for lengths between 1–50kb. Simulated reads were mapped to genome GRCh38. A read is considered as correctly mapping if the reported mapping interval has ≥ 10%, 40%, 70% overlap with the truth interval. *paftools.js mapeval* was used to evaluate the mapping accuracy.

while mapping accuracy at the 70% dropped below 0.90 and 0.97 for simulated CLR and ONT reads, respectively.

A common application of LRS in human genetics is detecting structural variation [2, 14]. We evaluated SV detection from mapped reads across combinations of technologies, aligners, and SV calling algorithms on both simulated datasets and real dataset. Reads were aligned with lra, pbmm2/minimap2, and ngmlr, and variants were detected with pbsv, Sniffles and cuteSV [2, 15, 16]. The pbmm2 method encapsulates minimap2 with technology and SV discovery specific parameters. The pbsv method did not run using ONT reads, and so pbsv analysis of ONT data is omitted. In the simulation study, we used SURVIVOR (https://github.com/fritzsedlazeck/SURVIVOR) to simulate insertions, deletions, inversions and more complicated nested SVs, including deletion-inversion-deletions (INVDEL) and inverted duplications (INVDUP). We simulated 195 insertions and deletions of lengths between 50–10000 bases, 97 inversions of lengths between 600–2000 bases and 100 INVDEL and INVDUP of lengths between 600–1000 bases respectively using the human genome GRCh37 chromosomes 20–22 as a reference. HiFi and CLR reads were simulated by PBSIM and ONT reads were simulated by alchemy2. Truvari (https://github.com/spiralgenetics/truvari) was used to benchmark the insertions, deletions and inversions callset. Both pbsv and cuteSV calling results show similar F1 score (>= 0.974, 0.979, 0.974) for lra, minimap2 and ngmlr alignments on insertions and deletions over all data types. All three aligners achieved comparable F1 scores (>= 0.97) on inversions from cuteSV calling results over all data types (S3 Table). There was a lower F1 score for variants detected by Sniffles on lra alignment of the simulated CLR reads, which is inconsistent with other two callers. Complicated nested SVs like INVDEL presented a greater challenge, particularly for lower accuracy reads. Using simulated HiFi reads, 98 of the INVDEL and 94 of the INVDUP were correctly called by lra, with all alignment methods detecting at least 93 variants in each data set. The majority of the INVDEL variants detected by lra alignments using CLR (79) data missed one deletion, while variant calls from minimap2 and ngmlr correctly detect 94 and 74 INVDEL variants, respectively. A different pattern existed for ONT data, where 1–2 correct calls were detected from minimap2/ngmlr, and 37 by lra. The INVDUP calls were successfully detected from lra and ngmlr alignments, and represented as insertions by minimap2. The lra alignments resulted in 99 and 100 calls from the CLR and ONT simulated reads, and ngmlr 98 calls from both datasets (S4 Table).

We used pbsv and Sniffles to detect variation on the HiFi, CLR, and ONT data; the cuteSV variant calls had lower accuracy on PacBio data across multiple alignment methods (S7 Table) and were excluded from analysis. The majority of SVs are under 500 bases [14] and are

**Table 2. Statistics of structural variant call sets.**

| | | | lra | | | pbmm2 | | | ngmlr | | |
|---|---|---|---|---|---|---|---|---|---|---|---|
| | | | Number of calls | Average call size | Total size | Number of calls | Average call size | Total size | Number of calls | Average call size | Total size |
| pbsv | CLR | INS | 13177 | 560 | 7.38M | 12828 | 523 | 6.72M | 11416 | 537 | 6.14M |
| | | DEL | 9148 | 688 | 6.30M | 8926 | 603 | 5.39M | 8569 | 594 | 5.10M |
| | HiFi | INS | 13133 | 577 | 7.58M | 12655 | 488 | 6.19M | 10342 | 506 | 5.24M |
| | | DEL | 9037 | 619 | 5.60M | 8784 | 583 | 5.12M | 8445 | 575 | 4.86M |
| Sniffles | CLR | INS | 16377 | 1074 | 17.59M | 11592 | 785 | 9.10M | 8318 | 600 | 5.00M |
| | | DEL | 10624 | 820 | 8.71M | 9195 | 578 | 5.32M | 8371 | 739 | 6.19M |
| | HiFi | INS | 12004 | 805 | 9.67M | 10902 | 459 | 5.01M | 9716 | 436 | 4.24M |
| | | DEL | 8883 | 1073 | 9.54M | 7907 | 430 | 3.40M | 7698 | 672 | 5.18M |
| | ONT | INS | 13052 | 1084 | 14.15M | 12103 | 488 | 5.91M | 10314 | 476 | 4.91M |
| | | DEL | 10129 | 752 | 7.63M | 9238 | 550 | 5.08M | 8286 | 637 | 5.28M |

Summary of SV callsets from different aligners (lra/pbmm2/ngmlr) + SV callers (pbsv/Sniffles).

spanned by LRS alignments. Consequently, variant calls with similar representations of gaps should have converging callsets, which may be confirmed by comparing SV calls from different combinations of algorithms. The most consistent variant calling was found using pbsv on both types of PacBio using lra and pbmm2 alignments. These callsets ranged from 21,439–22,325 SVs, with a difference of 2.6–3.3% between alignment algorithms and 0.7–1.5% between data types (Table 2). Compared to the pbsv calls, there was greater variation between callsets generated using Sniffles across both data types and alignment algorithms, with a range of 16,689–27,001 SVs per callset. There was less variation between the size of callsets of different data types and the same alignment algorithm (average difference of 624 across pbsv callsets), compared to different alignment algorithms applied to the same data (an average difference of 2,290 across pbsv callsets). The differences between callsets not due to gap placement were broadly measured by comparing callsets with a low-stringency filter: within 50% of length and at most 1kb apart. Using this definition of overlap, the callsets from PacBio data using pbsv were most similar with no more than 20% of calls unique to an alignment method. When stratifying PacBio variants unique to a dataset overlap with genomic features, 29–46% of variants overlap tandem repeats, and 15–56% overlap segmental duplications (SD) (S2 Table). This indicates that calls in tandem repeats may have different breakpoints and are difficult to compare using standard overlap approaches [13].

A greater call count may reflect more sensitive detection of SV, or simply fragmentation of variants. To assess the accuracy of variant callsets, we compared callsets against the GIAB Tier 1 calls [13] using Truvari. The lra and pbmm2 based callsets outperform ngmlr based calls in precision and recall on all data types (Table 3 and Fig 2). While the F1 scores are effectively equivalent between lra and pbmm2 based variant calls on HiFi (0.970 vs 0.968) and CLR (0.967 vs 0.967) using pbsv, there is a substantial improvement on calls on ONT data made by Sniffles (0.942 vs 0.910). This indicates that the greater call count on ONT data includes is at least partially attributed to increasing recall, particularly in larger (>500 base) insertions, without affecting precision on the high-quality regions that were ascertained.

While the F1 scores are nearly equivalent on PacBio data, the combination of algorithms may contribute to a more complete evaluation of an LRS genome. Calls from the lra and pbmm2 alignments contribute 83 and 43 unique calls, respectively, that were annotated as true positives in the HiFi/pbsv call sets, and 134, 133 unique calls from the CLR/pbsv call sets. The average length of the uniquely called true positive SV among these callsets is 1986bp,

**Table 3. Truvari classification of variant calls.**

| | pbsv | | | | | | Sniffles | | | | | | | | |
|---|---|---|---|---|---|---|---|---|---|---|---|---|---|---|---|
| | HiFi | | | CLR | | | HiFi | | | CLR | | | ONT | | |
| | lra | pbmm2 | ngmlr | lra | pbmm2 | ngmlr | lra | minimap2 | ngmlr | lra | minimap2 | ngmlr | lra | minimap2 | ngmlr |
| TP base | **9466** | 9408 | 8433 | **9438** | **9414** | 9076 | **8924** | 8700 | 8015 | **8969** | 8082 | 6533 | **9085** | 8887 | 8471 |
| TP call | **9506** | 9449 | 8500 | **9516** | **9507** | 9208 | **8964** | 8713 | 8028 | **9101** | 8147 | 6567 | **9102** | 8902 | 8488 |
| FP | 404 | **390** | 396 | 448 | **418** | 428 | 392 | **357** | 800 | 2023 | 1492 | **1298** | **550** | 940 | 815 |
| FN | **175** | 233 | 1208 | **203** | **227** | 565 | **717** | 941 | 1626 | **672** | 1559 | 3108 | **556** | 754 | 1170 |
| s/FN | 108 | 93 | 176 | 169 | 151 | 268 | 573 | 598 | 573 | 629 | 1423 | 2387 | 477 | 464 | 410 |
| precision | 0.959 | **0.96** | 0.955 | 0.955 | **0.958** | 0.956 | 0.958 | **0.961** | 0.909 | 0.818 | **0.845** | 0.835 | **0.943** | 0.904 | 0.912 |
| recall | **0.982** | 0.976 | 0.875 | **0.979** | **0.976** | 0.941 | **0.926** | 0.902 | 0.831 | **0.930** | 0.838 | 0.678 | **0.942** | 0.922 | 0.879 |
| F1 score | **0.970** | 0.968 | 0.913 | **0.967** | **0.967** | 0.952 | **0.942** | 0.931 | 0.869 | **0.871** | 0.842 | 0.748 | **0.943** | 0.91 | 0.895 |

Truvari comparisons between lra, pbmm2/minimap2 and ngmlr using the Genome in a Bottle benchmark SV set. Optimal results in each category are shown in bold. TP-base means true positive calls in the benchmark SV curation set, while TP-call means true positive calls in the SV set from each aligner. False positive means the number of non-matching calls from the SV set from each aligner. False negative means the number of non-matching calls from the SV curation set.

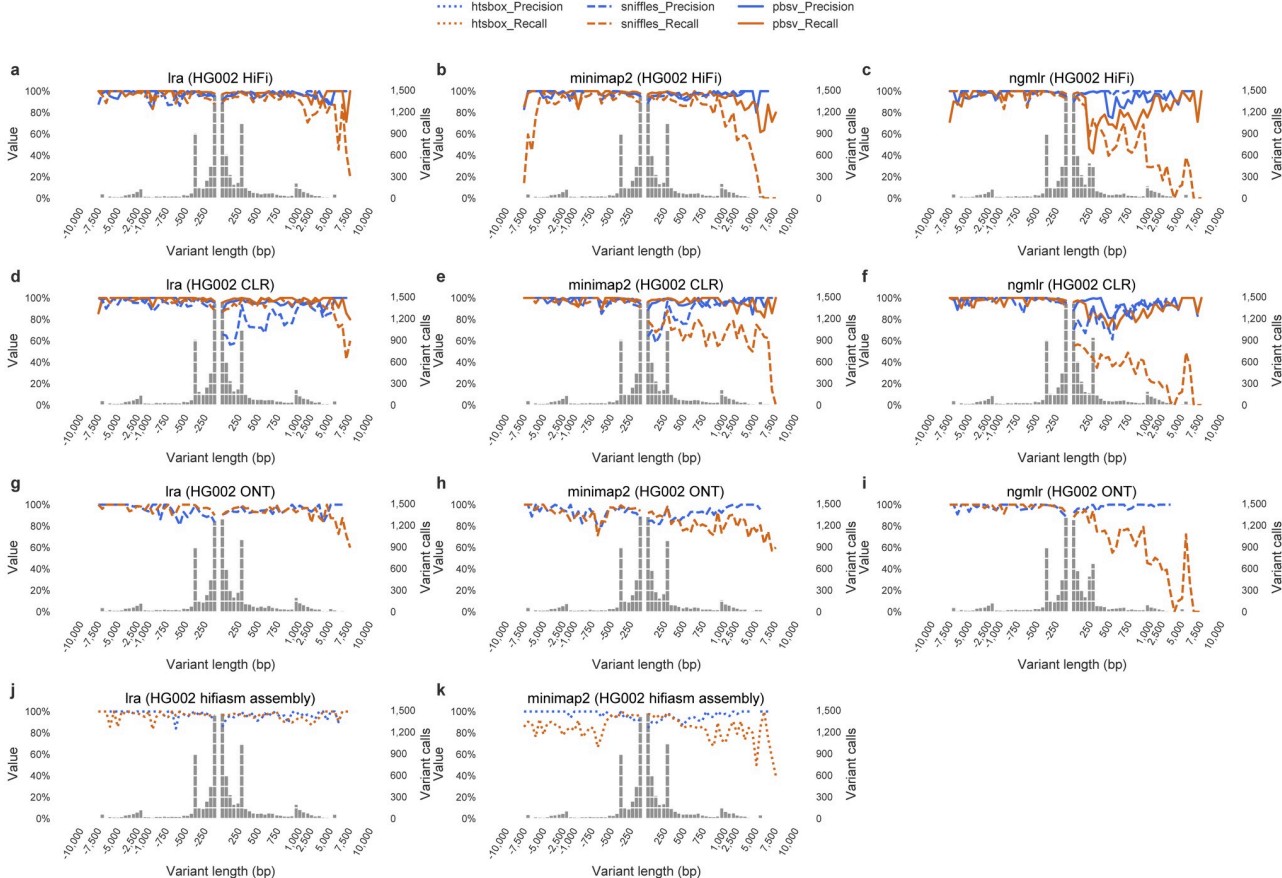

**Fig 2. The precision and recall of HG002 SV call sets.** The precision and recall of SV call sets from HG002 HiFi, CLR, ONT, hifiasm assembly, measured by Truvari using Genome in a Bottle benchmark SV callsets.

highlighting the challenge of detecting longer SV from read based alignments. To further assess the potential for callsets to be more comprehensive, benchmark variants annotated as false-negatives were compared to variants discovered directly from read alignments. Each call where at least 20% of the reads overlapping the call had an SV of the same type with the ratio of lengths between 0.6–1.4 was annotated as a supported false-negative (s/FN). Between 14.5%-83.2% of annotated false negatives from pbsv variant callsets, and 35.0%-93.6% of false-negative calls in Sniffles callsets were considered s/FN. Furthermore, there were 13% more supported false-negative calls in the PacBio lra/pbsv callsets compared to the pbmm2/pbsv callsets, possibly the result of tuning the pbsv variant discovery algorithm to the input generated by a particular alignment algorithm.

The accuracy of breakpoints was measured by comparing the boundaries of true positive SVs to the breakpoints of the GIAB truth data for each aligner/caller combination. We measured the percentage of SVs with perfect breakpoint boundaries and the average shifting distance between the left-most coordinate of SV boundaries. The average shifting distances of lra is 53bp across all data type and SV calling algorithms, compared to an average shifting distance of 46bp across all aligner/data/SV calling methods. The percentages of SVs with perfect breakpoints called from lra alignments ranged between 31–64% across all data and SV calling algorithms (S5 Table), which indicates in addition to alignment, the accuracy of breakpoints also depends on SV caller.

A comparison of callsets highlights how detecting inversions and SV in segmental duplications remains challenging. Inversion calls were generated by Sniffles. We filtered calls that overlapped the centromere. Each resulting callset was manually curated using dot-plots of the genome assemblies against the reference. Calls were classified as true positives if at least one haplotype demonstrated the inversion, an inverted duplication if the dot-plot signature was a fixed inverted duplication or inverted transposition, false-positive if both haplotypes did not show an inversion or an inverted duplication, and NA if not possible to validate with the assembly, commonly in pericentric regions. Across all data types, ngmlr based alignments discovered the most inversions and inverted duplications annotated as true-positives, with calls ranging from 8077–44,538 bases (Table 4). On average, 75 inversions and inverted duplicatoins were detected using ngmlr alignments, versus 61 for both minimap2 and lra, indicating that additional development may be required to accurately detect rearranged DNA with minimal computational burden.

When sequencing depth is sufficiently high coverage, comprehensive haplotype-resolved *de novo* assemblies can be used to detect variation instead of read alignments. Both lra and

**Table 4. Classification of inversions detected in HG002 using PacBio and ONT reads.**

| Dataset | Aligner | TP | Dup | NA | FP |
|---------|---------|----|-----|----|-----|
| HiFi | lra | 30 | 21 | 8 | 26 |
| | minimap2 | 59 | 4 | 4 | 8 |
| | ngmlr | 58 | 17 | 2 | 16 |
| CLR | lra | 65 | 9 | 9 | 36 |
| | minimap2 | 69 | 1 | 9 | 20 |
| | ngmlr | 66 | 17 | 9 | 17 |
| ONT | lra | 53 | 7 | 7 | 18 |
| | minimap2 | 44 | 7 | 1 | 6 |
| | ngmlr | 58 | 10 | 3 | 9 |

TP, true positive using manual curation. Dup, inverted duplication misclassified as an inversion on both haplotypes, NA not possible to manually curate, FP—no signature of inversion in read nor assembled haplotype dot-plots.

minimap2 were used to align contigs of a haplotype-resolved *de novo* assembly of HG002 constructed by the hifiasm assembler [10]. The haplotype assemblies resolve 2.97Gb and 3.07Gb, with N50 values 50.55 and 42.92 Mb. Both alignment methods have similar runtime to align *de novo* assembly contigs to GRCh37, although lra requires 1.5-fold the memory. Over 95% of each haplotype assembly was mapped by each method; 2.92/2.97 Gb were mapped by minimap2, and 2.87/2.97Gb mapped by lra. It was not possible to use ngmlr to align contigs. Variants were detected from the lra and minimap2 alignments using *htsbox pileup -cuf*. Calls from different haplotypes that overlap at least 30% were determined as homozygous calls and the rest were classified as heterozygous calls. Additionally, calls in centromere regions were removed for both minimap2 and lra callsets. This generated 25,982 calls by lra, and 26,341 by minimap2. Truvari analysis of the calls gives an F1 score of 0.955 by lra calls, and 0.933 for minimap2 calls (Table 5). There are 233 calls from the true-positive annotations of the lra HiFi/pbsv callset missing from the lra assembly callset. Upon inspection, the majority of these missed calls are due to a combination of the process used to merge haplotype calls into a diploid callset, shifted breakpoints relative to the GIAB annotation, and haplotype dropout/swapping of *de novo* assembly sequences.

Diversity studies are increasingly using SMS *de novo* assemblies to measure variation [17, 18] rather than from sequencing reads. An integrated analysis of the regions of the genome spanned by assembly alignments and the raw-read base calls from multiple alignment and SV discovery methods can help reveal the extent that variation is discovered and potentially missing from a high-quality haplotype-resolved *de novo* assembly. Across all read technologies, alignment algorithms, and variant discovery methods, 167 unique calls representing 0.6% of an average read-based SV callset were in regions not mapped by either haplotype of the assembly. Of these, 113 are in segmental duplications (SD). Because SDs make up roughly 5% of the human genome, the majority of SD regions that are mapped by SMS reads in this study are assembled in the hifiasm genome. The lra alignments detect 53 and 60 large SV (>20kb) totaling over 3Mb from the haplotype 1 and haplotype 2 assemblies that are not found in the read SV callsets (S2 Text). Because the length of SV negatively correlates with allele frequency, and is associated with enrichment in genome-wide association studies significant loci [19], these larger variants represent a class of variation that is biologically important for discovery.

**Table 5. Assembly based calls comparison.**

|  | HG002 HiFi hifiasm assembly | |
|---|---|---|
|  | **lra** | **minimap2** |
| Insertion, hom | 6276 | 6494 |
| Insertion, het | 8852 | 8783 |
| Deletion, hom | 3518 | 3895 |
| Deletion, het | 7336 | 7169 |
| TP base | **9310** | 9035 |
| TP call | **9412** | 9062 |
| FP | **546** | 692 |
| FN | **331** | 606 |
| precision | **0.945** | 0.929 |
| recall | **0.966** | 0.937 |
| F1 score | **0.955** | 0.933 |

*htsbox pileup -cuf* is used to call SVs from contig lra and minimap2 alignment. Calls from lra and minimap2 assembly alignments are classified as homozygous and heterozygous by comparing calls from two haplotypes. Truvari comparison results between these two callsets are shown.

**Table 6. Read based support of assembly calls.**

| Assembly | SV type | total | filtered | supported | fraction |
|---|---|---|---|---|---|
| Haplotype 1 lra | DEL | 6101 | 5874 | 5767 | 0.982 |
| Haplotype 1 minimap2 | DEL | 6078 | 5924 | 5841 | **0.986** |
| Haplotype 2 lra | DEL | 6287 | 6061 | 5963 | 0.984 |
| Haplotype 2 minimap2 | DEL | 6267 | 6107 | 6033 | **0.989** |
| Haplotype 1 lra | INS | 9736 | 9354 | 9275 | 0.992 |
| Haplotype 1 minimap2 | INS | 9882 | 9619 | 9587 | **0.997** |
| Haplotype 2 lra | INS | 10032 | 9651 | 9568 | 0.991 |
| Haplotype 2 minimap2 | INS | 10183 | 9890 | 9859 | **0.997** |

The two haplotype assemblies are considered separately to avoid complications of merging into a diploid callset. Calls are produced by lra and minimap2, with only deletion and insertion SV classes considered. The total calls are the original calls produced by each method, and filtered are calls excluding centromeres and 50kb of flanking sequence. The supported SV have at least four reads supporting the call from either lra or minimap2 alignments.

The high-confidence callset used in Truvari analysis contains fewer than 10,000 SVs, less than half what are expected to be found in a human genome. To gauge the specificity of assembly-based calls outside this set, SV contained in read alignments were compared to the SV discovered from assemblies. An SV detected from an individual read supported an assembly based SV if the call was the same type (e.g. insertion or deletion), was within 1kb, and the ratio of SV lengths was between 0.5–2. When using PacBio CLR aligned reads, nearly all (99.998–100%) of SV annotated by Truvari as true-positives had at least four supporting reads aligned by lra or minimap2. Using this approach, 98.3% of lra deletion SV, and 98.7% of minimap2 deletion SV are supported by reads, and 99.1%/99.7% of insertion SV from lra/minimap2, are supported by reads (Table 6). This is greater than the precision measured by Truvari on assembly-based calls: 0.955 for lra, and 0.933 for minimap2, indicating an underestimate of the callset precision. When inspecting the SV calls that are not supported by read alignments, many are due to differential placement of gaps causing disagreement of SV length between the read and assembly-based calls.

## Materials and methods

The alignment of reads and contigs to a reference are generally defined by the maximally scoring local alignment of a query $q$ to a set of target sequences collectively referred to as a target $t$ with a match bonus and penalties for mismatch, gaps, and inversions/translocations. lra uses a heuristic to find an approximate local alignment employing the commonly used seeding, chaining, and refinement approach that has been applied to all scales of alignment from short-read, long-read, and whole-genome alignment [3, 20, 21]. Each alignment proceeds in four broad steps: seed sequence matching, clustering, chaining, and finally alignment refinement.

Many recent advances have been made in sequence mapping using a subsampled index on a reference using minimizers or locally sensitive hashing [22, 23]. A minimizer index is parameterized by a k-mer size $k$ and window size $w$, and indexes the position of the lexicographically least canonical k-mer in every sequence of length $w$ across the genome. We develop a variant on the approach of minimizers that uses adaptive thresholds to limit the total number of positions sampled in unique regions of a genome, and increase the sampling of positions near paralog-specific variants that distinguish repetitive regions.

Optimal sets of matches between the query and reference are selected in two phases using CG-SDP. Given a set of fragments $\Phi = \{\alpha_1, \ldots, \alpha_n\}$, each fragment is defined by a start point

and an end point on a Cartesian plane and a weight: $\alpha_i = ((x_i^s, y_i^s), (x_i^e, y_i^e), l_i)$. The basic CG-SDP formulation is to define a chain of fragments $C \subset [1, \ldots, n]$ that maximizes

$$\sum_{j=1}^{|C|} l_{C_j} - \sum_{j=1}^{|C|-1} gap(\alpha_{C_j}, \alpha_{C_{j+1}}) \tag{1}$$

such that $\alpha_{C_{j+1}}$ is above and to the right of $\alpha_{C_j}$, and $gap$ is a concave function. An algorithm was presented for chaining using a concave gap cost model [7], however there are no alignment methods that implement this approach. This method has been implemented in lra and extended to allow for inversions in the optimal chain. Additionally, the original description of the algorithm requires asynchronous processing, which we have updated to use standard serial computation. The details of determining minimizers and the chaining algorithm are given below.

## Building a minimizer index and sequence matching

We add three additional parameters to generate a minimzer index: $F^M$, $N^M$ and $W^G$ that limit the density of minimizers that are selected. An initial set of minimizers is determined in the standard approach, with minimizer k-mer parameter $k$ and window $w$ [22]. Next, minimizers of multiplicity larger than $F^M$ are removed. Then the reference is partitioned into intervals of length $W^G$, and the remaining minimizers starting in each interval are selected in order of their multiplicity in the genome until the first $N^M$ minimizers are obtained, in a more simplified version of weighted minimizer sampling [24]. Different parameter settings are used to index a genome for different sequencing technologies and contig alignments. When indexing the genome for aligning HiFi reads, 867M minimizers from total 1015M were left after filtering by frequency threshold $F^M$. In total, 117M minimizers were selected after the final filtering based on $N^M$ and $W^G$. All minimizers from a query sequence are matched against the filtered set of minimizers from the reference. The result of the sequence matching is a set of anchors $A = \{\beta_1, \ldots, \beta_n\}$, where $\beta_i$ is a tuple $(x_i, y_j, k)$, where $q[x_i, x_{i+1}, \ldots, x_{i+k-1}]$ of the query matches $t[y_j, y_{j+1}, \ldots, y_{j+k-1}]$ of the target.

The chains of anchors for low-accuracy single-pass SMS reads may be sparse or have spurious off-diagonal matches in repetitive sequences. The detailed alignment may be calculated using dynamic programming within the region that is chained, however to limit the computation required for refining the alignment we implement an additional index of local minimizers that are used to refine an alignment once a coarse-chaining has been done. The local index is parameterized by $k^l < k, w^l < w$, and a tiling length $W^l$, and is composed of a collection of separate minimizer indexes for sequence intervals of length $W^l$ tiling across both the query and the target.

## Clustering

Although CG-SDP can be applied to all anchors $A$, for efficiency a greedy approach is used to cluster anchors that would likely be together on an optimal chain. These clusters may be used to filter out spurious matches in low accuracy reads, or may be chained directly on high accuracy reads and contig mapping. When forming alignments from chained clusters, it is necessary to have a cluster refining step that divides rough clusters into non-overlapping fine clusters to avoid chaining that skips biological variation in repetitive sequences.

**Rough clustering.** Rough clustering partitions anchors into clusters representing approximate intervals on the query and target that are aligned (Fig 3a), and serves to exclude noisy anchors unlikely to be chained in an alignment by CG-SDP. Denoting the forward diagonal of

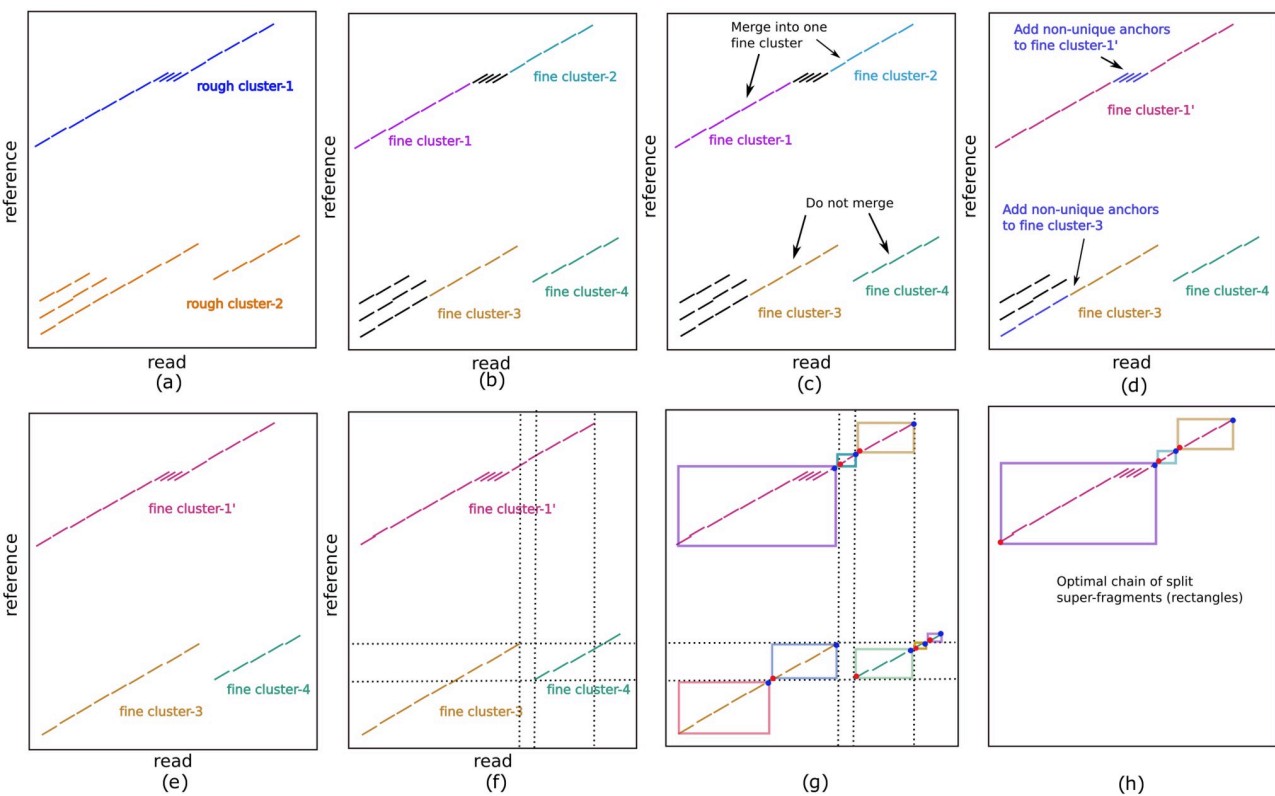

**Fig 3. Example of clustering anchors prior to CG-SDP. a**, Two rough clusters (blue and orange), which are far from each other on the reference. **b**, The initial four fine clusters defined from the contiguous stretches of unique anchors. **c**, fine cluster-1 and fine cluster-2 are merged because their diagonal difference is smaller than $D^F$ and projected distances between their endpoints is smaller than $G^{dist}$. fine cluster-3 and fine cluster-4 are not merged due to the large diagonal difference. **d**, Non-unique anchors in the trapezoid between fine cluster-1 and fine cluster-2 are added to the merged fine cluster-1, along with non-unique anchors in the trapezoid defined by the start of rough cluster-2 and the start of fine cluster-3. **e**, Three fine clusters are obtained after rough clustering and fine clustering. **f-h**, Splitting of overlapping fine-clusters: **f**, overlap of clusters. **g**, Boundaries of split clusters defined by a start (red dot) and an end (blue dot). **h**, the optimal chain of split super-fragments.

each anchor $\beta_i$ as $f_i = y_i - x_i$, and the reverse diagonal $r_i = x_i + y_i$, a sorted anchor order $O = [o_1, \ldots, o_n]$ is defined by ordering anchors by forward diagonal and then $x$ coordinate. A reverse sorted order $O^{rev} = [o_1^{rev}, \ldots, o_n^{rev}]$ is similarly defined sorting on reverse diagonal and $x$ coordinate. This will be used to detect alignments on the reverse strand, but because the operations are the same as on the forward strand, only subsequent steps using the forward sorted order are given. The set of rough clusters is defined by partitioning $O$ into non-overlapping intervals such that every anchor indexed in an interval has a diagonal within $D^R$ of the preceding anchor in the interval. Intervals are greedily assigned with first interval starting at the first index in $O$, and subsequent intervals starting on after a gap of more than $D^R$ between anchors. Intervals with few elements (defined by a *minClusterSize* parameter) are discarded, and the rough clusters $R = \{R_1, \ldots, R_{NR}\}$ are defined from the set of anchors included in each interval. The value of $D^R$ is chosen so that rough clusters are likely to contain at least *minClusterSize* true anchors from a read (default to 3 for CLR and ONT, and 10 for HiFi/contigs). For CLR and ONT data, we empirically determined a sufficient $D^R$ is 200, and 150 for HiFi and assembly contig alignment. For low accuracy reads, chains are formed by running CG-SDP on all matches retained in rough clustering. For high accuracy reads, the clusters must be post-processed with fine-clustering prior to being chained.

**Fine clustering.** For mapping hich accuracy reads, each rough cluster is processed independently by dividing into non overlapping fine clusters, where each fine cluster consists of anchors on a close diagonal $D^F$, with endpoints that do not overlap. The first step of CG-SDP will be applied to chain the fine clusters and find an approximate alignment between $q$ and $t$. Each fine cluster $C_j$ is defined by all of the anchors contained in the cluster, and endpoints $(x_j^s, y_j^s)$ and $(x_j^e, y_j^e)$, where $x_j^s, y_j^s$ are the minimum $x, y$ coordinates of the starting points of all the anchors in $C_j$, and $x_j^e, y_j^e$ are the maximum $x, y$ coordinates of the ending points of all the anchors in $C_j$. To define the fine clusters, anchors in each rough cluster are first sorted by Cartesian coordinate. Within each rough cluster, an anchor is defined as unique when the k-mer of the match is not repeated in the cluster. Fine clusters are initialized as runs of unique anchors in the Cartesian ordering that are on a close diagonal, and the distance between the end of one anchor and the start of the next is small (Fig 3b). Every pair of fine clusters $C_j$ and $C_k$ are merged if their endpoints have diagonal differences smaller than $D^F$ and are within Cartesian distance $G^{dist}$ (Fig 3c), and all non-unique anchors within the trapezoid defined by $[(x_j^e, y_j^e - D^F), (x_j^e, y_j^e + D^F), (x_k^s, y_k^s - D^F), (x_k^s, y_k^s + D^F)]$ are included into the merged fine cluster (Fig 3d and 3e). The remaining non-unique anchors that are not added into the fine cluster are discarded. In the step of fine clustering, we empirically found $D^F = 500$ was able to distinguish clustering anchors in different tandem repeats and allowed a sufficient number of repetitive anchors to be included in the fine clusters.

## Cluster splitting and chaining

Each fine cluster $C_j \in C$ defines a super-fragment $F_j$ that unlike the minimizer fragments which have starting and endpoints along the same diagonal, the endpoints of $C_j$ and may not be along a diagonal. The set of all such fragments is $F$, and an optimal chain of fragments will be defined using CG-SDP. However, considering the rectangles defined by the boundaries of each fragment, the CG-SDP algorithm only selects fragments with non-overlapping rectangles in the optimal chain. Due to the repetitive nature of genomes, this may result in erroneously skipped fragments. To account for this, fine clusters are split at overlapping boundaries (Fig 3g). The start and end coordinates of all fine clusters are stored in a set that is queried to find boundary points appearing in the range of each fine cluster. The coordinates of each split cluster are set according to the first/last anchors appearing after/before the boundary point. An optimal chain of fragments $F^{SDP} \in F$ with corresponding clusters $C^{SDP}$ are then found using CG-SDP on $F$.

## Cluster refinement

The optimal chain of super-fragments $C^{SDP}$ contains the anchors from which the optimal alignment will be defined. A pairwise alignment may be created using dynamic programming on the substrings between the minimizer matches, however for high-error rate reads matches are sparse and the length of substrings may be too large to efficiently compute, or have too large of a diagonal gap to use banded alignment. A second anchoring step using the local minimizer index is used to detect shorter and more dense anchors. The local minimizer index contains $\lceil \frac{|t|}{W^l} \rceil$ and $\lceil \frac{|q|}{W^l} \rceil$ separate minimizer indexes for the target and query sequences that index each tiling substring of length $W^l$, accounting for the relative positions of the substrings in each sequence. Every pair of substrings from $q$ and $t$ that have an anchor in $C^{OPT}$ are compared using their local minimizer indexes to generate a resulting set of anchors $A^{local}$. To reduce runtime of CG-SDP on $A^{local}$, anchors that are adjacent in Cartesian sorted order and on the same diagonal are merged. The length of any merged anchor is the difference from the last endpoint

to the first starting point. The resulting merged fragments are chained using CG-SDP, as described in section Problem statement of chaining, and the anchors on this chain are denoted as $A^{local-opt}$.

## Alignment refinement

Banded alignment is used to create a pairwise alignment between anchors in $A^{local-opt}$. We assume that large gaps between anchors may be modeled using an affine alignment that allows a single large penalty-free gap. Given two sequences $q^{local}$ and $t^{local}$, a match matrix $M$, a gap penalty $\delta$, and an alignment band $B$, assume that $|q^{local}| < |t^{local}| - B$. A lower-score matrix $S^{lower}$ is calculated for a typical banded alignment band $B$ and gap penalty $\delta$, e.g $S^{lower}_{i,j} = \max\{S^{lower}_{i-1,j-1} + M[q^{local}_i, t^{local}_j], S^{lower}_{i,j-1} - \delta, S^{lower}_{i-1,j} - \delta\}$ if $|i - j| \le B$, $-\infty$ otherwise. Next, an upper-diagonal matrix $S^{upper}_{i,j}$ is calculated that allows for a single transition from the lower matrix with banded alignment. Denoting the length of the $q^{local}$ and $t^{local}$ as $l^q$ and $l^t$:

$$S^{upper}_{i,j} = \begin{cases} \max\{S^{upper}_{i-1,j-1} + M[q^{local}_i, t^{local}_j], S^{upper}_{i,j-1} - \delta, S^{upper}_{i-1,j} - \delta\} & \text{if } i - j - (l^t - l^q) < B , \\ S^{lower}_{j,j+B} & \text{if } l^t - l^q - (j - i) = B , \quad (2) \\ -\infty & \text{otherwise} \end{cases}$$

## Problem statement of chaining

We define a set of fragments $\Phi = \{\alpha_1, \ldots, \alpha_n\}$. Each fragment $\alpha_i$ is associated with a lower left start point $(x^s_i, y^s_i)$ and upper right end point $(x^e_i, y^e_i)$, and a score $l_i$. For minimizer fragments, the upper right endpoints are a fixed distance from the lower points, e.g. $(x^s_i + k, y^s_i + k)$. For super fragments defined by fine clusters, the endpoints and starting points may not be on the same diagonal. The starting point of a fragment $\alpha$ is denoted $s(\alpha)$, and the end $e(\alpha)$. A point $(x_i, y_i)$ is above $(x_j, y_j)$ if $x_i > x_j, y_i > y_j$ (conversely $(x_j, y_j)$ is below $(x_i, y_i)$). The chaining score is defined by Eq 1 where $gap(\alpha_{C_j}, \alpha_{C_{j+1}})$ is a concave function of $|(y^s_{C_{j+1}} - x^s_{C_{j+1}}) - (y^e_{C_j} - x^e_{C_j})|$, the difference between the forward diagonals of the endpoint of $\alpha_{C_j}$ and the starting point of $\alpha_{Cj}+ 1$. The naive way to solve this problem takes $O(n^2)$ in time. By applying CG-SDP [7], the runtime is $O(n \log(n)^2)$. Below, the solution provided by Eppstein, Galil, and Giancarlo is described both with increase clarity from the original description, with a modification that enables calculation with synchronous computation. Finally, the method is extended to allow for inversions, similar to [25].

**SDP algorithm with concave gap cost function—Defining subproblems.** For simplicity, assume both sequences are the same length $l$ and that all points are in $[0, l)$ (e.g. shifted by the minimum $x$ and $y$ coordinate), and are on a $l \times l$ grid. To speed the chaining algorithm, the search for the fragment that precedes another on an optimal chain is divided into multiple overlapping subproblems that may be solved independently and more efficiently than the naive scan, and the globally optimal score for each point is selected from each of the subproblems that overlap it. Each subproblem divides a block of $t^{sub}$ columns or rows of the search space into an $A$-part and $B$-part covering $\lceil \frac{t^{sub}}{2} \rceil$ and $\lfloor \frac{t^{sub}}{2} \rfloor$ columns/rows respectively, where $A$-part contains all the endpoints and $B$-part contains all the startpoints in the corresponding columns/rows. When the size of a subproblem is only one column/row, the $A$-part of the corresponding subproblem is set to be empty. Each subproblem is described by the label $d \in \{column, row\}$, the starting and ending rows and columns of the subproblem $A.s, A.e, B.s, B.e$, and a set of lists collectively referenced as $DATA$ that are used in the calculation of optimal chains. The full set of

subproblems $Sub(d, s, e)$ are generated recursively as:

$$\begin{cases} \{(d, \emptyset, \emptyset, s, e, DATA)\} & \text{if } s == e, \\ \\ \{(d, s, \lfloor\frac{s+e}{2}\rfloor, \lfloor\frac{s+e}{2}\rfloor + 1, e, DATA) \cup Sub(d, s, \lfloor\frac{s+e}{2}\rfloor) \cup Sub(d, \lfloor\frac{s+e}{2}\rfloor + 1, e)\} & \text{otherwise} \end{cases} \quad (3)$$

The more detailed pseudocode for this partitioning is given in S1 Algorithm. Fig 4 visualizes the column and row subproblem division of a simple case with six fragments.

Each subproblem is processed by finding the optimal endpoint from the $A$-part that precedes each starting point in corresponding $B$-part. For a starting point $p_i$ that is assigned to the $B$-parts of a set of column-based subproblems, the union of the $A$-parts of those subproblems form the entire plane to the left of the point. Similarly for a starting point assigned to the $B$-part of a set of row subproblems, the union of the $A$-parts of those subproblems form the plane below the point. After solving for the optimal preceding endpoint in all of the subproblems in which $p_i$ is contained, the one with maximum score among these endpoints is the global optimal, which represents the optimal chain from all endpoints below and to the left of $p_i$.

Once the subproblems are defined, the list elements of $DATA$ are allocated and initialized. The following elements are associated with the $A$-part of a column/row subproblem:

- $D_I$: The forward diagonals in increasing/decreasing order overlapped by endpoints in the $A$-part.

- $D_V$: The optimal chaining score for diagonals overlapped by endpoints in $A$-part. $D_V[s]$ holds the optimal value of forward diagonal $D_I[s]$.

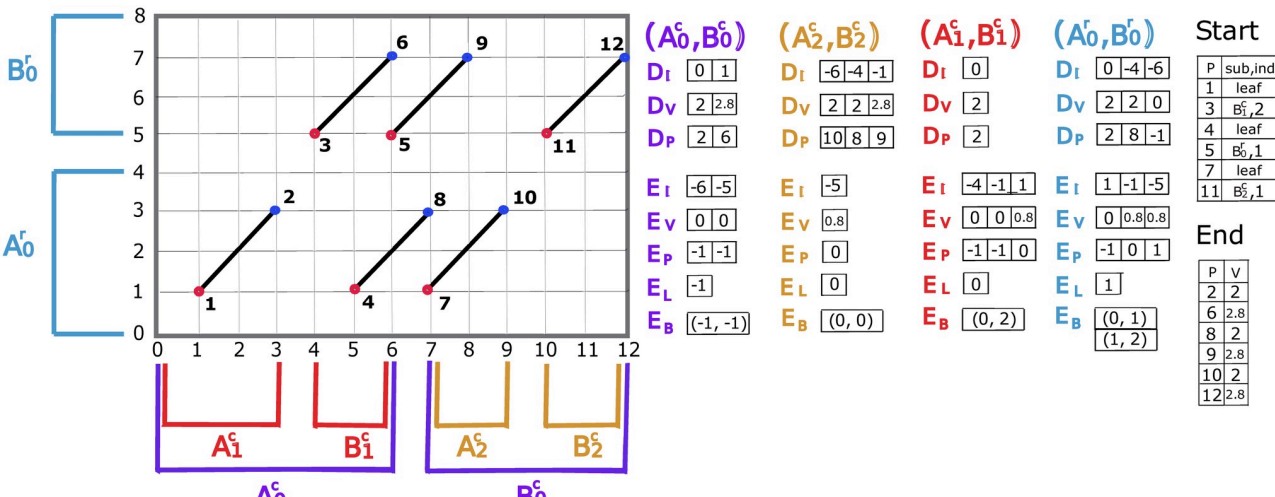

**Fig 4. Visualization of subproblems divisions.** The data structures needed for each subproblem: $D_I, D_V, D_P, E_I, E_V, E_P, E_L$ and $E_B$ after the processing of all the 12 points. Points are numbered in the order of processing (Cartesian sorted order). 12 points are assigned into three column subproblems $(A_0^c, B_0^c)$, $(A_1^c, B_1^c)$, $(A_2^c, B_2^c)$ and one row subproblem $(A_0^r, B_0^r)$, where starting points are assigned to $A$-part and endpoints are assigned to $B$-part. Leaf subproblems are not shown for simplicity. *Start* and *End* are used for the traceback of the optimal chain. *Start* stores $sub$—the index of the subproblem which yields the optimal chaining score up to a starting point and $ind$—$\varphi(E_I, f_i)$, where $f_i$ is the diagonal of the starting point. *End* stores the optimal value for each endpoint. For this toy example, the match bonus of every fragment is 2 and the gap cost function of appending fragment $\alpha_j$ to fragment $\alpha_i$ is $gap(\alpha_i, \alpha_j) = 0.25 * log(|(y_i^e - x_i^e) - (y_j^s - x_j^s) + 1|) + 1$, where $(x_i^e, y_i^e)$ is the endpoint of $\alpha_i$ and $(x_j^s, y_j^s)$ is the starting point of $\alpha_j$. *End* indicates that there are three optimal chains achieving the optimal value: 2.8. By tracing back, chain-1: point-1, point-2, point-3, point-6, chain-2: point-1, point-2, point-5, point-9 and chain-3: point-7, point-10, point-11, point-12.

- $D_P$: Index of point with optimal score along diagonal. $D_P[s]$ references the point with the best score of $D_V[s]$.

  The following elements are associated with the $B$-part of a column/row subproblem:

- $E_I$: The forward diagonals in increasing/decreasing order overlapped by starting points in the $B$-part.

- $E_V$: Similar to $D_V$, the locally best chaining value.

- $E_P$: Index of the forward diagonal in $D_I$ that leads to the best chaining value $E_V[s]$ for the forward diagonal $E_I[s]$.

- $E_B$: A block structure used in the calculation of $E_V$.

- $E_L$: Index of the most recent value in $D_V$ used to compute values in $E_V$.

  For each point, we have two lists that reference the subproblems a point is contained in:

- $SA$, A list of subproblems that contain a point that is in an $A$-part of a subproblem.

- $SB$, A list of subproblems that contain a point that is in a $B$-part of a subproblem

To make the description of the method more concise, the operation $\varphi(D_I, f)$ and $\varphi(E_I, f)$ are defined to give the index of diagonal $f$ in $D_I$ and $E_I$, respectively.

**SDP algorithm with concave gap cost function—Conquering subproblems.** The original description of CG-SDP, uses asynchronous processing during solving subproblems [7]. Here we give an alternative description of the algorithm, and provide an approach to solve subproblems in a way that allows synchronous processing of points in Cartesian order for a more simple implementation. The framework for solving a subproblem is first described, and then an order of processing points is given to solve for optimal fragment chaining.

For any subproblem, the optimal chains between endpoints in the $A$-part to starting points in the $B$-part are found using the $D_I$, $D_V$, $D_P$, $E_I$, $E_V$, $E_P$ arrays, variable $E_L$ and block list $E_B$. Consider two points $p_a$ in $A$-part, and $p_b$ in $B$-part that are on diagonals $f_a$ and $f_b$, respectively. An invariant for any subproblem is that if the optimal chain ending with the fragment that has endpoint $p_a$ has been solved and is referenced as $Score(p_a)$, the score of chaining $p_a$ with $p_b$ is $Score(p_a) + gap(|f_b - f_a|)$. Because the gap cost only depends on diagonal and not the coordinates of a point, all points in $A$-part on the same diagonal $f_a$ will have the same cost to chain with $p_b$. More generally, the score to chain any starting point in $B$-part to an ending point in $A$-part is equal to the score of chaining a diagonal in $B$ to a diagonal in $A$. The score of an optimal chain up to point $p_b$ in $B$-part, $E_V[j]$, where $j = \varphi(E_I, f_b)$, is found for column-based problems as:

$$E_V[j] = \max_{k:D_I[k] \leq E_I[j]} D_V[k] + gap(|E_I[j] - D_I[k]|) \tag{4}$$

For row-based subproblems:

$$E_V[j] = \max_{k:D_I[k] > E_I[j]} D_V[k] + gap(|E_I[j] - D_I[k]|) \tag{5}$$

The naive approach to solve for all values of $E_V$ scales quadratically as $O(|E_V||D_V|)$. A problem of this form:

$$E[j] = \min_{k \leq j} D[k] + w(k, j), \tag{6}$$

where $w$ is a concave function and $D[k]$ is a linear transformation of $E[k]$, was solved in $O(|D|$

$\log|E|)$ time using the auxiliary block data structure $E_B$ [26, 27]. In this solution, iteration is performed with one pass over the $D$ array. After each iteration $i$, the invariant holds: each element in $E$ has been set to reference which element of the prefix $D[1, \ldots, n]$ gives the minimum value for Eq (6). Efficiency is gained by not updating $E$ explicitly but by storing indexes in the data structure $E_B$ that may be updated in $\log(|E|)$ time by applying a function $Update(D[i], E_B)$ on each iteration. The data structure $E_B$ has an operation that can reconstruct $E$, $E[k] = \Omega(E_B, k)$. The details of $E_B$ and $Update$ may be derived from [27]. In the application to CG-SDP, $D$ is replaced by $D_V$ and $E$ by $E_V$, and Eqs (4) and (5) solve for the optimal chains between diagonals from an $A$-part of a subproblem to diagonals in the $B$-part of the column and row subproblem respectively. An important detail in our implementation is that in column-based problems an $Update$ operation only affects references in $E_V$ that are on a greater or equal diagonal than the current element in $D_V$, and for row-based problems elements in $E_V$ are only affected for diagonals that are less than the current element in $D_V$. However, it is not possible to apply Eqs (4) and (5) directly. Because points are processed in Cartesian order, but $D_V$ in forward diagonal order, values of $D_V$ are not solved in increasing order and not all values of $D_V$ are guaranteed to be solved when values of $E_V$ are computed. In [7] this is accounted for by asynchronous computation. Below, an approach is described to solve with a standard model of computation by calling $Update(D_V[i], E_B)$ using subsets of $D_V$ as they become known. Contrary to the customary model of divide and conquer, where subproblems are completely solved before combining into a global solution, this solves portions of subproblems on each iteration.

Points are processed in order of $x$ and then $y$ coordinate, determining the value of the optimal chain up to the current processed point. The value of a starting point is the value of the optimal chain that chains the starting point to an endpoint below it. When no endpoints are below a starting point, the value of this starting point is trivially set to 0. The value of an endpoint is simply the value of the optimal chain preceding the corresponding starting point plus the value of the fragment.

When processing an endpoint $p_i$, the starting point $p_s$ of the corresponding fragment $\alpha_i$ has been solved because points are processed in Cartesian order. The value of the chain at the endpoint is $Score(p_s) + l_i$, where $l_i$ is the match bonus of fragment $\alpha_i$. In order for $p_i$ to be used when solving for starting points that are above it, the value of $D_V$ must be updated in subproblems for which $p_i$ is a point in the $A$-part. The $SA$ list is used to determine which subproblems contain $p_i$ in a $A$-part. Suppose the forward diagonal of $p_i$ is $f_i$. In each subproblem in $SA$, $Score(p_i)$ is passed to $D_V[j]$ and $D_V[j]$ is set to $\max\{Score(p_i), D_V[j]\}$, where $j = \varphi(D_I, f_i)$, and $D_P[j]$ is set to $i$ if $D_V[j]$ gets updated.

When processing a starting point $p_i$, the optimal value must be calculated in each of the $B$-parts of subproblems that include $p_i$, and the global optimal value will be selected among them. For any subproblem, this can be achieved by solving for $E_V[j]$ using Eq (4) for column-based or Eq (5) for row-based subproblems, where $j = \varphi(E_I, f_i)$ and $f_i$ is the diagonal of $p_i$. For a column-based subproblem, this requires that the values of $D_V[k]$ have been solved where $D_I[k] \leq f_i$, and for row-based subproblems, the values of $D_V[k]$ must have been solved where $D_I[k] > f_i$. These correspond diagonals overlapping points that fall in the region below and to the left of the $p_i$ and are exactly what have been processed when solving for points in Cartesian sorted order. Thus all required values in $D_V$ arrays (though not the entire $D_V$ array) are available when solving for $p_i$.

The value of $E_V[j]$ is optimal once $Update(D_V[k], E_B)$ has been called on all diagonals that contain potential predecessors to points on diagonal $E_I[j]$. The function $Update(D_V[k], E_B)$ must be called only once per element in $D_V$ and in increasing order. However because points are processed in Cartesian sorted order, values of $E_V$ are solved in arbitrary order, and calling $Update(D_V[k], E_B)$ can reference elements in $D_V$ multiple times. To account for this, we

maintain an index $E_L$ for each subproblem that keeps track of the last element of $D_V$ which has been processed by $Update(D_V[k], E_B)$. Before solving any points, $E_L$ is initialized to -1 in every subproblem, and when processing points will be assigned to reference the most recently updated diagonal from $D_V$. When processing starting point $p_i$ in a column-based subproblem, $E_V[j]$ must be solved, where $j = \varphi(E_I, f_i)$. If $D_I[E_L] > f_i$, $Update$ has been called on all values of $D_V$ that $E_V[j]$ relies on, and the state of $E_B$ contains the optimal value for $E_V[j]$ such that it may be calculated immediately from $\Omega(E_B, j)$. Otherwise, $Update(D_V[k], E_B)$ is called for $E_L < k < C$, where $D_I[C]$ is the first diagonal from the left that is larger than $f_i$ in array $D_I$, and $E_V[j]$ may be calculated from $\Omega(E_B, j)$. Similarly, for row-based subproblems if $f_i \geq D_I[E_L]$, the value of $E_V[j]$ may be assigned immediately from $\Omega(E_B, j)$ where $j = \varphi(E_I, f_i)$. Otherwise, $Update(D_V[k], E_B)$ is called for $E_L < k < C$, where $D_I[C]$ is the first diagonal from the left that is smaller than or equal to $f_i$ in array $D_I$, and $E_V[j]$ can be retrieved from $\Omega(E_B, j)$. By comparing values $E_V[j]$ from all the subproblems in the $SB$ list of startpoint $p_i$, the maximum value for $p_i$ will be obtained and stored. The pseudocode and detailed example of solving points and conquering subproblems are given in the S2 Algorithm and S3 Fig.

**Extension to inversion cases.**   This extension is inspired by the work [25]. As explained in the previous section, when chaining fragments in forward direction, two points—a lower left start point $(x_i^{s1}, y_i^{s1})$ and a upper right endpoint $(x_i^{e1}, y_i^{e1})$ would be associated to each fragment $\alpha_i$. $(x_i^{s1}, y_i^{s1})$ can be chained to an endpoint below and to the left of it and $(x_i^{e1}, y_i^{e1})$ can be the predecessor of a starting point above and to the right of it. To allow inversions to happen, fragments must be allowed to be chained in the reverse direction. To account for this, we associate two more points to each fragment $\alpha_i$, that are a upper left start point $(x_i^{s2}, y_i^{s2})$ and a lower right endpoint $(x_i^{e2}, y_i^{e2})$. The start point $(x_i^{s2}, y_i^{s2})$ can only be chained to endpoint $(x_j^{e2}, y_j^{e2})$ of some other fragment $\alpha_j$ that satisfies $x_j^{e2} < x_i^{s2}, y_j^{e2} > y_i^{s2}$. The endpoint $(x_i^{e2}, y_i^{e2})$ can precede a starting point $(x_k^{s2}, y_k^{s2})$ of some other fragment $\alpha_k$ that satisfies $x_k^{s2} > x_i^{e2}, y_k^{s2} < y_i^{e2}$. $(x_i^{s1}, y_i^{s1})$, $(x_i^{e1}, y_i^{e1})$, $(x_i^{s2}, y_i^{s2})$ and $(x_i^{e2}, y_i^{e2})$ can be represented as $s1(i)$, $e1(i)$, $s2(i)$ and $e2(i)$ respectively. Chaining fragment $\alpha_i$ to fragment $\alpha_j$ in the reverse direction equals to chaining the start point $s2(\alpha_i)$ to the endpoint $e2(\alpha_j)$. The cost of appending $s2(\alpha_i)$ to $s2(\alpha_j)$ in the reverse direction is $gap^{rev}(\alpha_j, \alpha_i)$, which is a concave function of $|(x_j^{e2} + y_j^{e2}) - (x_i^{s2} + y_i^{s2})|$, the difference between the reverse diagonals of fragments $\alpha_i$ and $\alpha_j$. In a short word, $s2(\alpha)$ and $e2(\alpha)$ of each fragment $\alpha$ would be responsible for the possible chaining of $\alpha$ in the reverse direction, while $s1(\alpha)$ and $e1(\alpha)$ would be responsible for the forward direction.

We used the same column and row subproblem dividing scheme to sort all $s1$ and $e1$ points and assign them into column-1 and row-1 subproblems. Then all $s2$ and $e2$ points are sorted and assigned to column-2 and row-2 subproblems in the same way. Arrays $D_I, D_V, D_P, E_I, E_V, E_P$, the block structure $E_B$ and variable $E_L$ are allocated and initialized for each subproblem. Lists $SA_1$ and $SA_2$ reference column-1/row-1 and column-2/row-2 subproblems that each endpoint $e1$ and $e2$ is in respectively. Similarly, lists $SB_1$ and $SB_2$ references column-1/row-1 and column-2/row-2 subproblems that starting points $s1$ and $s2$ in respectively. The only difference between column-2/row-2 and column-1/row-1 subproblems is that column-2/row-2 stores reverse diagonal instead of forward diagonal in $D_I$ and $E_I$ arrays.

The steps to solve subproblems to allow chaining in both forward and reverse directions are highly similar to section SDP algorithm with concave gap cost function—conquering subproblems. Points are processed in order of $x$ and then $y$ coordinate. When a starting point $s1(\alpha_i)$ is being processed, $E_V[j]$, where $j = \varphi(E_I, f_i)$ and $f_i$ is the forward diagonal, will be computed from column-1/row-1 subproblems in $SB_1$. The maximum $E_V[j]$ of those subproblems is the value of optimal chain up to $s1(\alpha_i)$, $Score(s1(\alpha_i))$, where $s1(\alpha_i)$ is forwardly chained to an endpoint $e1$ $(\alpha_j)$ below and to the left of it. Similarly, when a starting point $s2(\alpha_i)$ is being processed, $E_V[j]$,

where $j = \varphi(E_I, r_i)$ and $r_i$ is the reverse diagonal, will be computed from column-2/row-2 subproblems in $SB_2$. The maximum $E_V[j]$ of those subproblems is the value of optimal chain up to $s2(\alpha_i)$, $Score(s2(\alpha_i))$, where $s2(\alpha_i)$ is reversely chained to an endpoint $e2(\alpha_k)$ above and to the left of it.

After solving $s1(\alpha_i)$ and $s2(\alpha_i)$ for fragment $\alpha_i$, the value of the optimal chain up to fragment $\alpha_i$ can be calculated as $Score(\alpha_i) = \max\{Score(s1(\alpha_i)), Score(s2(\alpha_i))\} + l(\alpha_i)$, where $l(\alpha_i)$ is the match bonus of fragment $\alpha_i$. This optimal chain is chosen from all the possible chains that fragment $\alpha_i$ is forwardly or reversely chained to the predecessor in the left. When solving $e1(\alpha_i)$, $Score(\alpha_i)$, will be passed to array $D_V[j]$, where $j = \varphi(E_I, f_i)$ and $f_i$ is the forward diagonal, to column-1/row-1 subproblems in $SA_1$. And $D_P[j]$ will be updated to the index of point $e1(\alpha_i)$, if $Score(\alpha_i) > D_V[j]$. Similarly, when $e2(\alpha_i)$ is being processed, $Score(\alpha_i)$ will be passed to array $D_V$ of column-2/row-2 subproblems in $SA_2$ and $D_P$ will be updated.

Therefore, the addition of two points $s2(\alpha)$ and $e2(\alpha)$ for each fragment $\alpha$ make it possible to allow $\alpha$ to be chained in the reverse direction. Meanwhile, the overall time complexity and storage remain bounded by $O(n(\log(n)^2))$, where $n$ is the total number of fragments.

**Time complexity.** Assume there are $n$ fragments in total, list $SB/SA$ contains $O(\log(n))$ subproblems that a point is in. In section SDP algorithm with concave gap cost function—conquering subproblems, we mention that when a starting point $p_i$ with forward diagonal $f_i$ is being processed, $Update(D_V[k], E_B)$ is called for $E_L < k < C$, where $D_I[C]$ is the first diagonal from the left that is larger than/smaller than or equal to $f_i$ in column/row subproblems. And $E_V[j]$ can be retrieved in $O(\log(n))$ time from the block structure $E_B$ by calling $\Omega(E_B, j)$, where $j = \varphi(E_I, f_i)$. Procedure $Update$ may be called several times for a starting point in each subproblem. In order to make it easy to quantify the total time complexity of $Update$ procedures, we consider that each $Update$ procedure is called right after $D_V[j]$ is updated by some endpoint $p_i$ with diagonal $f_i$, where $j = \varphi(E_I, f_i)$. When an endpoint is being solved, there are $O(\log(n))$ subproblems associated with it and in each subproblem $Update$ takes $O(\log(n))$ to conduct. Therefore, it takes $O((\log(n))^2)$ time to solve the subproblems that are associated with an endpoint. When a starting point is being processed, there are $O(\log(n))$ subproblems it is in and in each subproblem $E_V[j]$ can be computed from the block structure $E_B$ in $O(\log(n))$ time. Therefore, it takes $O((\log(n))^2)$ time to tackle subproblems that are associated with a starting point. Since there are $n$ fragments in total, the time complexity of processing all the points and subproblems is bounded by $O(n \log(n)^2)$.

## Discussion

The initial description of CG-SDP was given in 1992 in two publications, one covering an affine-cost gap function, and another with a concave-cost gap function [7, 28]. While there have been many implementations of affine cost SDP, no sequence alignment methods have been implemented using SDP with a concave-cost gap function. In the original description of the algorithm, the processing of a starting point is blocked until all subproblems the point relies on are solved, and then the process is unblocked and processing resumes. We find that this blocking and unblocking strategy is not necessary, and the addition of data structures to keep track of the state of computation of subproblems enables solving the problem with a standard model of computation. We used two additional strategies to effectively employ SDP in lra: an iterative refinement process where a large number of anchors from the initial minimizer search are grouped into a small super-fragments that are chained using SDP, and once a rough alignment has been found a new set of matches with smaller anchors is calculated using the local miminizer indexes. As a result, alignment is of similar speed to state of the art algorithms, without the need for single-instruction multiple-data (SIMD) processing; runtime was slower

when using an SIMD alignment library [29] possibly due to the overhead of invoking the library functions.

The lra alignments have competitive runtime and memory usage compared to minimap2. Using two different SV discovery algorithms, pbsv and Sniffles, we show it is possible to use lra alignments to discover SV using PacBio HiFi, CLR, and Oxford Nanopore reads, as well as directly from aligned *de novo* assembly contigs. The performance for SV detection using PacBio reads and the pbsv algorithm is similar between lra and minimap2, with lra demonstrating a small gain in recall over larger SV events. Importantly, the availability of multiple alignment algorithms can help improve the results of studies that require high sensitivity and specificity, such as Mendelian analysis. The greater improvement in SV discovery metrics on ONT data aligned by lra additionally highlights the utility of using multiple algorithms to analyze sequencing data.

Finally, as *de novo* assembly of genomes becomes more routine, it is important to have accurate methods of SV detection from contig alignments. We used lra to align a haplotype-resolved genome assembled from PacBio HiFi reads and detect SV. When compared to the GIAB SV benchmark data, the lra alignments show slightly lower precision and recall than read-based SV detection. However, when the assembly-based SV are compared to SV detected in an orthogonal read dataset, nearly all ($> 98.8\%$) variants discovered from assemblies are supported by read alignments. This indicates few differences between the HiFi based assembly and the reference are due to assembly error, and the annotated precision of the callset is likely an underestimate. A variant callset is effectively a list of operations that may be applied to the reference genome to reconstruct a sample genome. Because the HiFi assembly has few assembly errors on the same size scale as an SV, this supports the development of an alternative model for validating SV callsets in which the reconstructed genome is compared to the haplotype-resolved assembly, rather than by comparing callsets. This may be used to validate calls inside the high-confidence regions defined where a haplotype-resolved assembly has been confidently generated.

## Supporting information

**S1 Fig. The distribution of read lengths from the HG002 HiFi, CLR, and ONT data.** (PDF)

**S2 Fig. The cumulative number of bases from the HG002 HiFi, CLR, and ONT data.** (PDF)

**S3 Fig. A detailed example of visualizing subproblems division.** The data structures for each subproblem: $D_I, D_V, D_P, E_I, E_V, E_P, E_L, E_B$ and the process of subproblems solving. The horizontal axis represents the query, while the vertical axis represents the target. Points are numbered in Cartesian sorted order, which is the processing order. 12 points are assigned into three column subproblems $(A_0^c, B_0^c), (A_1^c, B_1^c), (A_2^c, B_2^c)$ and one row subproblem $(A_0^r, B_0^r)$, where starting points are assigned to $A$-part and endpoints are assigned to $B$-part. Leaf subproblems are not shown for simplicity. Start and End are used for the trace-back of the optimal chain. *Start* stores *sub*—the index of the subproblem which yields the optimal chaining score up to a starting point and *ind*—the index of $f_i$ in array $E_I$, that is $\varphi(E_I, f_i)$, where $f_i$ is the diagonal of the starting point. *End* stores the optimal value for each endpoint. For this toy example, gap cost of appending fragment $\alpha_j$ to fragment $\alpha_i$ is $gap(\alpha_i, \alpha_j) = 0.25 * log(|(y_i^e - x_i^e) - (y_j^s - x_j^s) + 1|) + 1$, where $(x_i^e, y_i^e)$ is the endpoint of $\alpha_i$ and $(x_j^s, y_j^s)$ is the startpoint of $\alpha_j$. **m**, shows the regions that each subproblem covers and the initialized data structures for each subproblem. There are of three column subproblems and one row subproblems (leaf subproblems are not shown for

simplicity): $(A_0^c, B_0^c)$, $(A_1^c, B_1^c)$, $(A_2^c, B_2^c)$ and $(A_0^r, B_0^r)$. **a-l** shows how the data structures of subproblems that are associated with the point being processed in each step are updated. Note that entries that are updated are highlighted by orange. **a**, shows for $startpoint - 1$, it is a leaf subproblem that yields the value of the optimal chain up to $startpoint - 1$. **b**, shows when processing $endpoint - 2$, the optimal value up to it is $Score(startpoint - 1) + 2$, where 2 is the match bonus of the fragment. Array $End$ is then updated. Since $endpoint - 2$ is in $A_0^c, A_1^c, A_0^r$, the corresponding $D_V$ entries are updated to 2 and corresponding $D_P$ entries update to the index of $endpoint - 2$. **c**, shows when processing $startpoint - 3$, it is in the $B$-parts of subproblems $(A_1^c, B_1^c)$ and $(A_0^r, B_0^r)$. $startpoint - 3$ is located in $E_I[2]$ of $(A_1^c, B_1^c)$, so $Update(D_V[0], E_B)$ would be called to get the value of $E_V[2]$. The purple color highlighting shows what forward diagonals in $E_V$ would be updated by $D_V[0]$. $E_P[2]$ would be updated to point to $D_I[0]$. $startpoint - 3$ is located in $E_I[2]$ of $(A_0^r, B_0^r)$ and no forward diagonals in $D_I$ used to update $E_V[2]$. Therefore, in $Start$, $sub$ and $ind$ for $startpoint - 3$ are updated to $B_1^c$, 2. **d**, shows when processing $startpoint - 4$, it is in $B_1^c$ and locates in $E_I[0]$ of $(A_1^c, B_1^c)$. Since there is no forward diagonal in $D_I$ can be used to update $E_V[0]$, it is a leaf subproblem that yields the optimal chaining value up to $startpoint - 4$ in $Start$. **e**, shows when processing $startpoint - 5$, it is in $B_1^c$ and $B_0^r$. In $(A_1^c, B_1^c)$, there is no forward diagonal can be used to update $E_V[1]$. In $(A_0^r, B_0^r)$, $Update(D_V[0], E_B)$ is called to update the block structure $E_B$, so $E_V[1]$ and $E_V[2]$ would be computed from $E_B$. In $Start$, $sub$ and $ind$ for $startpoint - 5$ are updated to $B_0^r$, 1. **f**, **g**, **h**, **i**, **j**, **k**, **l**: show the subproblems solving and data structures updating for the rest of points. After processing all 12 points, three optimal chains can be obtained by tracing back, which are $chain - 1 = [startpoint - 1, endpoint - 2, startpoint - 3, endpoint - 6]$, $chain - 2 = [startpoint - 1, endpoint - 2, startpoint - 5, endpoint - 9]$ and $chain - 3 = [startpoint - 7, endpoint - 10, startpoint - 11, endpoint - 12]$.
(PDF)

**S1 Text. Impact of the local minimizer index.**
(PDF)

**S2 Text. Assembly exclusive calls.**
(PDF)

**S1 Algorithm. Defining subproblems.**
(PDF)

**S2 Algorithm. Sparse dynamic programming with convex gap cost.**
(PDF)

**S1 Table. Shared variant calls.** Considering HiFi/CLR + pbsv callsets and ONT + Sniffles callsets, the size of the intersection of every two callsets are shown. The intersection is calculated using *Truvari bench* on two called vcfs.
(PDF)

**S2 Table. Calls unique to a callset in tandem repeats and segmental duplications. Entry with pbmm2 as row and lra as column means the number of unique calls in pbmm2 callset when comparing pbmm2 callset and lra callset.**
(PDF)

**S3 Table. Comparison of the Truvari result between all combinations of aligners and SV callers on simulated HiFi, CLR and ONT dataset with simulated SVs.** SVs: Indels (insertions and deletions), inversions were simulated by SUVIVOR. All SVs were simulated by SUVIVOR. HiFi and CLR reads were simulated by PBSIM and ONT reads were simulated by alchemy2. We simulated 195 Indels (insertions and deletions) of lengths between 50-10000

bases, 97 inversions of lengths between 600-2000 bases.
(PDF)

**S4 Table. Comparison of the Truvari result between all combinations of aligners and SV callers on simulated HiFi, CLR and ONT dataset with complicated nested SVs.** Complicated nested SVs: deletion-inversion-deletions (INVDEL), inverted-duplications (INVDUP), which were simulated by SUVIVOR. HiFi and CLR reads were simulated by PBSIM and ONT reads were simulated by alchemy2, which is distributed with lra source. We simulated 100 deletion-inversion-deletions and inverted-duplications of lengths between 600-1000 bases respectively. Inversion-deletion is a type of nested SV where an inversion is flanked by 2 deletions and inversion-duplication means the duplicated sequence is inverted. For deletion-inversion-deletions, the inversion and two deletions have all been found in order to be counted as a TP. For cases where the inversion and only one flanked deletion are found were counted as Partial. For inverted-duplications, both the inversion and duplication need to be found in order to be counted as a TP. We found that minimap2 alignment find inverted-duplications as insertions, therefore, we didn't count that as TP.
(PDF)

**S5 Table. Comparison of the breakpoints on real datasets.** The breakpoints accuracy analysis was conducted by comparing the boundaries of true positive SVs from Truvari result to the curated SVs' breakpoints for each aligner/caller combination. Breakpoints accuracy is measured by the percentage of SVs with perfect breakpoint boundaries and the average shifting distance between the left-most coordinate of SV boundaries.
(PDF)

**S6 Table. Comparison of the breakpoints on simulated SVs.** The breakpoints accuracy analysis was conducted by comparing the boundaries of true positive SVs to the bondaries of ground truth SV for each aligner/caller combination. Breakpoints accuracy is measured by the percentage of SVs with perfect breakpoint boundaries and the average shifting distance between the left-most coordinate of SV boundaries.
(PDF)

**S7 Table. Truvari classification of cuteSV variant calls.** Truvari comparisons between lra, minimap2 and ngmlr using the Genome in a Bottle benchmark SV set. Optimal results in each category are shown in bold. TP-base means true positive calls in the benchmark SV curation set, while TP-call means true positive calls in the SV set from each aligner. False positive means the number of non-matching calls from the SV set from each aligner. False negative means the number of non-matching calls from the SV curation set.
(PDF)

## Acknowledgments

We appreciate the help from Peter Audano to test and fix various issues of lra. We thank lra users for suggesting features and pointing out issues.

## Author Contributions

**Conceptualization:** Jingwen Ren, Mark J. P. Chaisson.

**Data curation:** Jingwen Ren.

**Formal analysis:** Jingwen Ren, Mark J. P. Chaisson.

**Funding acquisition:** Mark J. P. Chaisson.

**Investigation:** Jingwen Ren, Mark J. P. Chaisson.

**Methodology:** Jingwen Ren, Mark J. P. Chaisson.

**Project administration:** Mark J. P. Chaisson.

**Software:** Jingwen Ren, Mark J. P. Chaisson.

**Supervision:** Mark J. P. Chaisson.

**Validation:** Jingwen Ren.

**Writing – original draft:** Jingwen Ren, Mark J. P. Chaisson.

**Writing – review & editing:** Jingwen Ren, Mark J. P. Chaisson.

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
