## [Decision Letter · Decision Letter 0]

25 Jan 2021

Dear Dr. Chaisson,

Thank you very much for submitting your manuscript "lra: A long read aligner for sequences and contigs" for consideration at PLOS Computational Biology.

As with all papers reviewed by the journal, your manuscript was reviewed by members of the editorial board and by several independent reviewers. In light of the reviews (below this email), we would like to invite the resubmission of a significantly-revised version that takes into account the reviewers' comments. The most critical points that need to be fully addressed are further evaluation of mapping accuracy, concerns about high false positive rates and improving the documentation of the tool to highlight its benefits.

We cannot make any decision about publication until we have seen the revised manuscript and your response to the reviewers' comments. Your revised manuscript is also likely to be sent to reviewers for further evaluation.

Sincerely,

Ferhat Ay, Ph.D

Associate Editor

PLOS Computational Biology

Jian Ma

Deputy Editor

PLOS Computational Biology

Reviewer's Responses to Questions

**Comments to the Authors:**

Reviewer #1: Ren & Chaisson described lra, a new long-read aligner specialized for calling structural variations (SVs). Lra implements a chaining algorithm with concave gap cost functions. Although the algorithm is not new, as the authors said, this is probably the first time the algorithm gets implemented. This manuscript represents an important addition in this field. Critical observations are as follows:

1) I tried lra v1.1.0 on reads simulated by PBSIM. At mapping quality 60, lra wrongly mapped 1.3% of reads. This is not as good as NGMLR. While simulation based evaluation may not reflect the accuracy on real data, it is a good sanity check and indicative of low-level algorithmic issues. I wonder if the authors can improve the mapping accuracy further. The file is available at https://github.com/lh3/minimap2/releases/download/v2.0/pb-1.fa.gz and the mapping result can be evaluated by "paftools.js mapeval -r 1e-6".

2) For a second test, I mapped the HG002 maternal hifiasm assembly to GRCh38 and call variants with "htsbox -vcf ref.fa -q5 -S10000 -T20 srt.bam | awk '$6==1'". Minimap2 -xasm5 calls 3.75M variants, which is about right. Lra calls 9.53M. The number is too high, suggesting the majority are false positives. I guess this is caused by inaccurate base alignment. I understand the authors are focusing on SVs, but the high false positive rate of small variants is concerning. It would be good if the authors can try to improve that.

3) For each query sequence, minimap2 chains all anchors in one round and produces multiple chains. In comparison, lra clusters anchors first and then chains in each cluster. While lra chaining may be theoretically better, clustering is heuristic and may not be as good. Combining clustering and chaining might hurt overall mapping accuracy. The authors said they use clustering "for efficiency". How much efficiency is affected if we use lra chaining for all anchors?

Minor comments

4) Minimap2+paftools is primarily developed for calling small variants, not for calling SVs. We can increase the recall of SV calling by adding options "-r2k -z2000,200" or even "-r7k -z2000,200". This trades performance and small variant accuracy for SV accuracy.

5) The manuscript is using concave gap cost functions, not convex.

6) Apparently, lra requires C++14 or C++17 to compile. However, Makefile is not using an option like -std=c++14. I got a compilation error.

Reviewer: Heng Li

Reviewer #2: Long read sequencing is a technology that becomes more important in recent days. Alignment of long reads is the fundamental step for many downstream analyses. In this manuscript, the authors described the design of lra, and compared the output to other programs. The tool is extreme valuable for long reads data analyses. It is challenging to understand all details described in the manuscript, but the paper is well-written, solid and with enough details. I also tested the program. It is easy to install and use. For an actively emerging area, any new tools are welcomed, and lra is one of the excellent ones from my point of view. Thus, I suggest accepting the manuscript with some minor revisions as shown below. I suggest the authors to improve the GitHub page to make it easier for users to understand the benefit of using lra.

Minor points:

Page 2 line 47, change “translocation” to “translocations”.

Page 5 line 150, change “section 3.3” to something else, as currently there is no title like “3.3”. The same for other similar cases.

There are a lot of abbreviations in this paper, such as “DATA” (line 206), “SA”, “SB” (line 237). It is better to add some explanations to make it easier for readers to follow. SA means subproblem A part?

Line 343, change “can be only be” to “can only be”.

Line 413, not a problem but I am surprised that GRCh37 instead of GRCh38 is used. Maybe because the curated variant file is based on GRCh37? Why It is “mapped to GRCh38” in line 427?

Line 423, “the number of bases aligned by lra and minimap2 are within 0.03-7%”. I do not understand. Do you mean the difference between the alignments by the two programs?

**Have all data underlying the figures and results presented in the manuscript been provided?**

Reviewer #1: Yes

Reviewer #2: Yes

PLOS authors have the option to publish the peer review history of their article (what does this mean?). If published, this will include your full peer review and any attached files.

Reviewer #1: **Yes: **Heng Li

Reviewer #2: No
---

## [Decision Letter · Decision Letter 1]

13 May 2021

Dear Chaisson,

We are pleased to inform you that your manuscript 'lra: A long read aligner for sequences and contigs' has been provisionally accepted for publication in PLOS Computational Biology.

Also, we strongly encourage you to address the few remaining points raised by two reviewers. Since these are very minor, we did not want to make you go through another round of revisions but rather address these together with other editorial edits needed in one final submission.  

Best regards,

Ferhat Ay, Ph.D

Associate Editor

PLOS Computational Biology

Jian Ma

Deputy Editor

PLOS Computational Biology

Reviewer's Responses to Questions

**Comments to the Authors:**

Reviewer #1: I have tried the latest LRA from github on the T2T CHM13 assembly and a HG00733 assembly. LRA crashed on both inputs. Nonetheless, based on the alignments it has outputted, I can see that LRA has greatly improved the alignment quality, which has addressed most of my major concerns in the previous round. Please fix the bug and make sure LRA runs under addressSanitizer. I also have a couple of minor comments. These are just suggestions that the authors may ignore:

1) It would be good to use a command-line option parser that is compatible with POSIX getopt. This will be more convenient to users and is less error prone.

2) In the SAM output, we use "=" for sequence matches and "X" for mismatches. We don't mix "M" and "X" together.

Reviewer #2: The authors have answered all my questions and included a great number of improvements compared to last submission. One tiny flaw may be “when a 40% overlap with the simulated interval is required for a correct alignment (Figure ??)”, where the meaning of “??” is unknown. The authors may consider adding some subtitles for “3 Results”. Otherwise, I think the manuscript is in great shape.

Reviewer #3: I have gone through the revision, and the authors have adequately

addressed all my comments. I have no further comments.

**Have the authors made all data and (if applicable) computational code underlying the findings in their manuscript fully available?**

Reviewer #1: Yes

Reviewer #2: Yes

Reviewer #3: Yes

PLOS authors have the option to publish the peer review history of their article (what does this mean?). If published, this will include your full peer review and any attached files.

Reviewer #1: **Yes: **Heng Li

Reviewer #2: No

Reviewer #3: No

---

## [Editor Report · Acceptance letter]

16 Jun 2021

PCOMPBIOL-D-20-02184R1 

lra: A long read aligner for sequences and contigs

Dear Dr Chaisson,

I am pleased to inform you that your manuscript has been formally accepted for publication in PLOS Computational Biology. Your manuscript is now with our production department and you will be notified of the publication date in due course.

With kind regards,

Zsofi Zombor
